# Evaluation of Haematological and Immunological Parameters of the ASFV Lv17/WB/Rie1 Strain and Its Derived Mutant Lv17/WB/Rie1/d110-11L against ASFV Challenge Infection in Domestic Pigs

**DOI:** 10.3390/vaccines11071277

**Published:** 2023-07-24

**Authors:** Giulia Franzoni, Stefano Petrini, István Mészáros, Silvia Dei Giudici, Cecilia Righi, Ferenc Olasz, Susanna Zinellu, Vivien Tamás, Michela Pela, Carmina Gallardo, Zoltán Zádori, Annalisa Oggiano, Francesco Feliziani

**Affiliations:** 1Department of Animal Health, Istituto Zooprofilattico Sperimentale della Sardegna, 07100 Sassari, Italy; silvia.deigiudici@izs-sardegna.it (S.D.G.); susanna.zinellu@izs-sardegna.it (S.Z.); annalisa.oggiano@izs-sardegna.it (A.O.); 2National Reference Laboratory for African Swine Fever, Istituto Zooprofilattico Sperimentale Umbria-Marche “Togo Rosati”, 06126 Perugia, Italy; c.righi@izsum.it (C.R.); m.pela@izsum.it (M.P.); f.feliziani@izsum.it (F.F.); 3Institute for Veterinary Medical Research, Hungária krt. 21, 1143 Budapest, Hungary; meszaros.istvan@vmri.hu (I.M.); olasz.ferenc@vmri.hu (F.O.); tamas.vivien@vmri.hu (V.T.); zadori.zoltan@vmri.hu (Z.Z.); 4Centro de Investigación en Sanidad Animal, CISA, INIA-CSIC, European Union Reference Laboratory for African Swine Fever (EURL), Valdeolmos, 28130 Madrid, Spain; gallardo@inia.csic.es

**Keywords:** ASFV, live-attenuated vaccine, domestic pigs, haematological parameters, cytokines

## Abstract

African swine fever virus (ASFV) is the etiological agent of a haemorrhagic disease that threatens the global pig industry. There is an urgency to develop a safe and efficient vaccine, but the knowledge of the immune–pathogenetic mechanisms behind ASFV infection is still very limited. In this paper, we evaluated the haematological and immunological parameters of domestic pigs vaccinated with the ASFV Lv17/WB/Rie1 strain or its derived mutant Lv17/WB/Rie1/d110-11L and then challenged with virulent Armenia/07 ASFV. Circulating levels of C-reactive protein (CRP), 13 key cytokines and 11 haematological parameters were evaluated throughout the study. Lv17/WB/Rie1 triggered an inflammatory response, with increased levels of CRP and pro-inflammatory cytokines, and induced lymphopenia, thrombocytopenia and a decline in red blood cell (RBC) parameters, although this was transitory. Lv17/WB/Rie1/d110-11L triggered only transitory thrombocytopenia and a mild inflammatory reaction, with no increase in serum levels of pro-inflammatory cytokines, but it raised IL-1Ra levels. Both strains counteracted several adverse reactions elicited by virulent challenge, like thrombocytopenia, a decline in RBC parameters, and inflammation. Within this paper, we provided a deep portrayal of the impact of diverse ASFV strains on the domestic pig’s immune system. A better understanding of these immune–pathological mechanisms would help to design suitable vaccines against this disease.

## 1. Introduction

ASF is a hemorrhagic disease of domestic pigs and wild boars, whose spread has reached pandemic proportions [1]. Its etiological agent is the ASF virus (ASFV), a large, enveloped, double-stranded DNA virus belonging to the *Asfarviridae* family [2]. The disease is currently present in Africa, Europe, Asia, and Oceania, and recent outbreaks have also been reported in the Americas and Italy [3,4]. The available control measures have failed to eliminate the disease in most ASF-affected countries [5]. Thus, there is a need to create a safe and efficient vaccine to avoid economic losses associated with ASF outbreaks. To date, several attempts have been made to create a vaccine against this threatening virus. Subunit vaccines or inactivated viruses provided advantages in terms of safety, but they could not confer protection to challenge infection [6,7]. Naturally attenuated field strains, such as OURT 88/3 or NH/P68, were able to confer satisfactory levels of protection to challenge infection with virulent homologous strains but maintained some residual virulence in some of the immunized domestic pigs [8,9]. Recently, several groups have tried to develop safe and efficient ASF live-attenuated vaccines (LAVs) [6,7,8,9].

ASF LAV strains have been obtained following different approaches, such as deleting genes associated with virulence from highly virulent or naturally attenuated field strains (such as NH/P68 or OURT 88/3) [8]. Several attempts were unfortunately unsuccessful; the resulting recombinant deleted vaccines either retained their virulence or they lost their protective efficacy [8,9]. 

A non-haemadsorbing (non-HAD) ASFV strain belonging to genotype II was isolated from a hunted wild boar in Latvia in 2017. This strain, later named Lv17/WB/Rie1, was characterized by an attenuated phenotype. An in vivo experiment in domestic pigs revealed that it induced only a subclinical form of ASFV [10]. Most interestingly, two months post-infection, immunized pigs were fully protected when infection was challenged with a virulent HAD genotype II ASFV [10]. It was later reported that Lv17/WB/Rie1 induced a less-severe disease compared to the other two HAD ASFV strains (Pol16/DP/OUT21 and Est16/WB/Viru8)—domestic pigs infected with these non-HAD strains developed a milder form of the disease [11]. Recent in vivo experiments in wild boar revealed that this non-HAD strain was relatively safe: animals immunized orally with a low dose developed only a slight transient fever after vaccination and booster; many tested wild boars orally immunized with a high dose of Lv17/WB/Rie1 survived against challenge infection [12]. These factors suggested that this vaccine prototype was a promising tool for the control of ASFV in wild boar. 

The safety of Lv17/WB/Rie1 might be further improved by the deletion of genes associated with virulence, but the obtained deleted mutant should retain the ability to protect immunized animals from challenge infection with ASF virus.

We recently reported that the MGF 110-11L gene was deleted from the Lv17/WB/Rie1 strain using the CRISPR/Cas9 method, and the harmlessness and efficacy of the candidate vaccine were tested in domestic pigs. The deleted Lv17/WB/Rie1/d110-11L vaccine showed reduced pathogenicity compared to the parental strain (Lv17/WB/Rie1) and induced protective immunity in vaccinated animals, although several mild clinical signs were observed [13]. We hypothesized that the Lv17/WB/Rie1/d110-11L vaccine candidate could evoke a diverse response of blood haematological and immunological parameters compared to its parental strain. To test our hypothesis, we aimed to assess the concentration of different serum cytokines after the administration of Lv17/WB/Ried1 and its deletion Lv17/WB/Rie1/d110-11L counterpart after experimental infection by the virulent Armenia/07 of the ASF virus in pigs. To date, little is known about the mechanisms underlying ASFV protection [14,15]. Indeed, a better understanding of virus–host interactions is needed to better design safe and effective ASFV vaccines.

## 2. Materials and Methods

### 2.1. Ethical Statement

The in vivo experiment was performed in Biosafety level 3 (BSL-3) facilities at the Istituto Zooprofilattico Umbria-Marche (IZSUM), Perugia, Italy. Thirteen cross-bred pigs were used in the study. Animals were purchased from a commercial farm and were acclimatized for seven days before starting the experiments. All animals were tested for ASFV antibodies using a commercial ELISA test (Ingezim PPA Compact K3 Ingenasa, Madrid, Spain) to prove their virus-free status. Animals were fed twice a day with a diet for fattening pigs and had access to water ad libitum [13]. The in vivo experiments were conducted under European legislation on the protection of animals used for scientific purposes. The experiments were carried out with the authorization of the Italian Ministry of Health (no. 424/2020-PR). 

### 2.2. Animals Experiments 

The experimental design is described in the manuscript published by Tamas et al., [13]. In brief, thirteen 3-month-old castrated crossbred pigs (Danish Landrace x Danish Suroc) were used in the experiments [13]. The domestic pigs were clinically healthy and were an average size of 30 Kg at the start of experiments. Animals were vaccinated with Lv17/WB/Rie1 (#6, #7, #8, #9, #10) or its derived mutant Lv17/WB/Rie1/d110-11L (#1, #2, #3, #4, #5), alongside a control group (#11, #12, #13) [13]. Blood samples collected in EDTA were used to assess complete blood cell counts. In parallel, whole blood without anticoagulant was collected to investigate serum levels of C-reactive protein (CRP) and of various cytokines. The above surveys were performed 0, 7, 14, and 21 days post-vaccination (dpv) and 7, and 14 days post-challenge (dpc) (Figure 1). Serum samples were stored at −80 °C until CRP and circulating cytokine levels were analyzed.

### 2.3. Collection of Blood Samples

Approximately 7 mL of blood (either EDTA-blood and whole-blood samples) was collected from each pig’s jugular vein. Disposable needles and vacutainer tubes were used for all animals (Kima, Padova, Italy). The EDTA-blood samples were used to evaluate changes in main blood parameters (see Section 2.4), whereas whole blood was used to monitor changes in CRP and cytokines levels (see Section 2.5). Serum samples were stored at −80 °C until the analysis of CRP and circulating cytokines levels. 

### 2.4. Complete Blood Count 

A complete blood count (CBC) was performed on swine EDTA blood samples. The samples were analyzed within two hours of collection. The main blood parameters were evaluated using a hematology analyzer (EosBIO, Italy) [16]. We reported the number of total white blood cells (1 × 10^3^/μL) and then divided them into neutrophils, lymphocytes, and monocytes. We analyzed the number of platelets (1 × 10^3^/μL) and their size (mean platelet volume, MPV) (fL). We reported the number of red blood cells (RBCs) (10^6^/µL), hematocrit (HCT), the volume percentage (%) of red blood cells in blood, and the amount of hemoglobin (HGB) expressed in grams per deciliter. Lastly, the mean corpuscular hemoglobin (MCH) and its concentration (MCHC) were evaluated. The list of parameters and their reference range are reported in Table 1. The reference values used for the analysis were previously validated in pigs [17,18,19,20].

### 2.5. Evaluation of C-Reactive Protein and Cytokines Levels in Serum Samples

Whole-blood samples were collected over time during the in vivo animal experiment. The blood samples were centrifuged at 850× *g* for 10 min, and serum was collected and stored at −80 °C until analysis. CRP serum levels were investigated using a commercial kit (Life Diagnostics Inc., West Chester, PA, USA), according to the manufacturer instructions. Sera were diluted to 1:2500 in a dilution buffer, and absorbance was read using an Epoch microplate reader (BioTek, Winoosky, VT, USA), as previously described [20]. In addition, sera samples were used to evaluate the levels of key cytokines: IL-1α, IL-1β, IL-1Ra, IL-2, IL-4, IL-6, IL-8, IL-10, IL-18, IFN-γ, GM-CSF, TNF. 

To determine the quantification of the cytokines (except IFN-β), a Porcine Cytokine/Chemokine Magnetic Bead Panel Multiplex assay (Merck Millipore, Darmstadt, Germany) and a Bioplex MAGPIX Multiplex Reader (Bio-Rad, Hercules, CA, USA) were used, according to the manufacturer’s instructions, as previously described [21,22]. IFN-β was instead assessed using a singleplex ELISA: IFN-β ELISA kit (MyBiosource, San Diego, CA, USA), following the manufacturer’s instruction. Absorbance was read using an Epoch microplate reader (BioTek, Winoosky, VT, USA), as previously described [22].

### 2.6. Statistical Analysis

The Shapiro–Wilk test was used to test the normal distribution of each independent variable. Data were then graphically and statistically analyzed with GraphPad Prism 9.01 (GraphPad Software Inc., La Jolla, CA, USA). Differences between immunized and control pigs were evaluated using an ANOVA followed by Dunnett’s multiple comparison tests or a Kruskal–Wallis test followed by Dunn’s multiple comparison test. The level of *p* < 0.05 was considered statistically significant.

## 3. Results

### 3.1. Complete Blood Count after Vaccination and Challenge of Pigs

A total of 11 haematological parameters were collected at different times during in vivo experimentation for a comprehensive evaluation of the vaccination/challenge’s impact on animal health status. 

First, circulating levels of platelet, red blood cells, and related parameters were investigated at different times post-vaccination and post-challenge. Vaccination with Lv17/WB/Rie1 and its derived mutant led to a decrease in platelet levels: statistically significant differences with the control group were observed at 7 dpv. Nevertheless, platelet levels remained within the reference range (100–900 10^3^/μL) (Figure 2). Post-challenge, the tested subject #11 (non-vaccinated) presented lower platelet levels than the reference range, and it died soon after sampling at 14 dpc. After the challenge with Armenia/07, none of the immunized subjects, either with Lv17/WB/Rie1 or its derived mutant Lv17/WB/Rie1/d110-11L, displayed platelet levels below reference ranges (PLV < 100 × 10^3^/μL). We observed differences between control and vaccinated pigs at 7 dpc but not at 14 dpc. Only two subjects in the control group were tested at 14 dpc, so it cannot be excluded that further statistically significant differences would have been detected by analyzing a greater number of pigs (Figure 2). No modulation of mean platelet volume (MPV) was observed either post-vaccination or post-challenge (Figure 2). 

In addition, IM administration of Lv17/WB/Rie1 led to a decrease in RBC numbers and an alteration in RBC parameters: statistically significant differences with the control group were observed in several parameters both at 7 dpv (HGB, HCT) and at 14 dpv (RBC number, HGB, HCT). Some vaccinated pigs (#6, #9, #10) had HGB and HCT levels below the reference ranges at 7 and/or 14 dpv (HGB < 10 g/dL, HCT < 32 × 10^3^/μL). Two of these three animals (#6, #10) died at 19 dpv. No difference between the control group and the attenuated strain Lv17/WB/Rie1/d110-11L was observed at 7, 14 and 21 dpv (Figure 3). We observed a marked decrease in RBC numbers, HGB and HCT in pigs belonging to the control group in the post-challenge period. In particular, tested subject #11 displayed these parameters below the reference range (RBC < 5 × 10^6^/μL) and it died at 14 dpc, soon after sampling. On the contrary, none of the vaccinated subjects, either with Lv17/WB/Rie1 or its derived mutant Lv17/WB/Rie1/d110-11L, presented post-challenge levels of RBC numbers, HGB, HCT below the reference range. Statistically significant differences between the control group and the Lv17/WB/Rie1/d110-11L group were observed at 7 dpc (RBC numbers) and 14 dpc (RBC numbers, HGB, HCT). In contrast, statistically significant differences between the control and Lv17/WB/Rie1 groups were detected only at 14 dpc (RBC numbers, HCT) (Figure 3). No modulation of MCH or MCHC were observed either post-immunization or post-challenge, with the exception of slightly higher levels of MCHC in controls compared to vaccinated pigs (with either Lv17/WB/Rie1 or its derived mutant Lv17/WB/Rie1/d110-11L) at both 7 dpc and 14 dpc (Figure 3).

The total number of leukocytes, then divided into granulocytes, lymphocytes, and monocytes, was monitored throughout the study (Figure 4). Vaccinated and control pigs did not present differences in circulating leukocytes, granulocytes, and monocytes at any tested time points. Between the control and the Lv17/WB/Rie1/d110-11L groups, we observed only a decreasing tendency (*p* = 0.0887) in the monocyte levels at 14 dpc. However, only two subjects in the control group were tested at 14 dpc, so it is not to be excluded that statistically significant differences would have been detected by analyzing a greater number of subjects. 

However, significant changes in lymphocyte levels were detected. As reported in Figure 4, a decrease in the lymphocyte levels was observed after IM administration of Lv17/WB/Rie1: differences with the control group were statistically significant at 7 dpv, while a tendency (*p* = 0.0789) was observed at 14 dpv. Some animals immunized with the non-haemoabsorbent strain had lymphocyte levels below the reference ranges (lymphocytes < 4.5 × 10^3^/μL): tested subjects #6 and #7 at 7 dpv and tested subjects #10 at 14 dpv. Two pigs (#6 and #10) died at 19 dpv. No differences between the control and Lv17/WB/Rie1/d110-11L group were observed at 7, 14, and 21 dpv. In this group, none of the subjects had lymphocyte levels below the reference range. Post-challenge, the control pig #11 presented levels of lymphocytes (but also total leukocytes) much lower than the reference range and indeed it died at 14 dpc, immediately after the sampling. On the contrary, none of the vaccinated subjects, either with Lv17/WB/Rie1 or its derived mutant Lv17/WB/Rie1/d110-11L, presented post-challenge lymphocyte levels outside the reference range (4.5–13 × 10^3^/μL) (Figure 4). A statistically significant difference in lymphocyte levels was observed at 14 dpc between the control and Lv17/WB/Rie1/d110-11L groups. Still, this was not the case in the Lv17/WB/Rie1 group; however, only two animals in the control group were tested at 14 dpc, so it cannot be excluded that differences would have been detected by analyzing a more significant number of subjects (Figure 4).

### 3.2. C-Reactive Protein Levels after Immunization and Challenge of Pigs

Serum levels of the acute phase inflammatory protein CRP were monitored during the study. Our results showed increased levels of this inflammatory marker in pigs vaccinated with either Lv17/WB/Rie1/d110-11L or parental Lv17/WB/Rie1, starting from 7 dpv (Figure 5). At the time of challenge (21 dpv), animals vaccinated with Lv17/WB/Rie1 presented higher levels of this protein compared to controls, whereas at that time point no statistically significant differences were observed between control and Lv17/WB/Rie1/d110-11L groups. Challenge with Armenia/07 resulted in increased levels of CRP in non-immunized pigs. After challenge, non-immunized animals (control group) presented higher levels of this protein compared to vaccinated pigs, with statistical significance for the Lv17/WB/Rie1/d110-11L group (Figure 5).

### 3.3. Evaluation of Cytokine Levels in Serum Samples

Finally, serum levels of 13 cytokines were determined at different times post-immunization and post-challenge for comprehensive immunological analysis.

IL-1α, IL-1β, IL-2, IL-6, IL-18, and TNF (formerly known as TNF-α) are pro-inflammatory cytokines [23], and their serum levels were monitored through the study. Our results showed increased levels of pro-inflammatory cytokines in pigs vaccinated with Lv17/WB/Rie1 compared to controls at both 7 dpv (IL-6) and 14 dpv (IL-1α, IL-1β, IL-2, IL-6) (Figure 6). At the time of challenge (21 dpv), no statistically significant differences were observed between control and immunized pigs. At 7 dpc, pigs vaccinated with Lv17/WB/Rie1/d110-11L presented lower levels of IL-1α, IL-1β, and IL-6 compared to the control group, although with statistical significance only for IL-1β, IL-6; no differences were detected between control and Lv17/WB/Rie1 groups (Figure 6). Differences between tested subjects of the Lv17/WB/Rie1 group were also observed. Some vaccinated pigs (#6, #7, #10) presented higher levels of IL-1α, IL-1β, IL-2, IL-6 compared to others of the same group (#8, #9). Interestingly, two of these three animals (#6, #10) died at 19 dpv. No differences between control and vaccinated groups were observed in the levels of IL-18 or TNF at any tested time points.

In parallel, the serum levels of two anti-inflammatory cytokines were monitored: IL-10 and IL-1Ra. IL-1Ra is a member of the IL-1 family, which counteracts the pro-inflammatory action of both IL-1α and IL-1β. It is a receptor antagonist; it competes for the same receptor of IL-1α and IL-1β (IL-1R), but its binding does not trigger any pro-inflammatory response [23]. Our results showed increased levels of IL-1Ra in pigs immunized with both Lv17/WB/Rie1/d110-11L and Lv17/WB/Rie1 at both 7 and 14 dpv (Figure 7). At the time of challenge (21 dpv), animals vaccinated with Lv17/WB/Rie1 presented higher levels of this cytokine compared to controls, whereas at that time, no statistically significant differences were observed between the control and the attenuated mutant Lv17/WB/Rie1/d110-11L groups. Fourteen days post-challenge with Armenia/07, increased levels of IL-1Ra were observed in non-vaccinated pigs compared to immunized pigs, although without statistical significance (Figure 7). Our results showed an increase in anti-inflammatory IL-10 in pigs immunized with Lv17/WB/Rie1 compared to controls at 7 and 14 dpv, although without statistical significance. Among Lv17/WB/Rie1-vaccinated pigs, tested subjects #6, #7, and #10 were those presenting higher levels of IL-10 before challenge. Interestingly, two of these three animals (#6, #10) died at 19 dpv. In addition, at 7 dpv, lower levels of IL-10 were observed for Lv17/WB/Rie1/d110-11L compared to controls (Figure 7). After the challenge, we did not observe increased levels of these cytokines in non-vaccinated pigs. Nevertheless, this might be linked to the lower number of pigs that survived infection: at 14 dpc, only two pigs from the control group were analyzed.

Vaccination with Lv17/WB/Rie1 did not result in the modulation of serum levels of IFN-β. On the contrary, increased levels of this cytokine were observed in pigs immunized with its derived mutant Lv17/WB/Rie1/d110-11L, although only in three out of five tested subjects and at different time points (Figure 8). No increased IFN-β levels were observed after the challenge in controls and Lv17/WB/Rie1-infected pigs (Figure 8). IM administration of Lv17/WB/Rie1 or its derived mutant Lv17/WB/Rie1/d110-11L did not result in increased serum levels of IFN-γ. In addition, levels of this cytokine were lower in pigs vaccinated with Lv17/WB/Rie1 compared to controls at 7, 14, and 21 dpv (Figure 8). Despite there being no increased levels of type II IFN in the sera of vaccinated pigs, three out of five pigs immunized Lv17/WB/Rie1 and all pigs vaccinated with its deleted mutant survived the challenge with Armenia07.

Immunization with both Lv17/WB/Rie1 and its deleted mutant did not mediate changes in serum levels of either GM-CSF, IL-4, or IL-8. No differences between control and vaccinated pigs were observed after the challenge with Armenia/07 either.

## 4. Discussion

ASFV is spreading fast around the world and there is an urgent need to develop a safe and effective vaccine to control ASF outbreak [6,7,8,9].

Lv17/WB/Rie1 seems a promising tool for the control of ASFV in wild boar [12], although some studies reported undesirable side effects in some of the vaccinated animals [10,12]. To overcome this safety issue, a new candidate vaccine was created [13] by deleting the gene MGF110-11L associated with virulence using the CRISPR/Cas9 method. As stated above, this candidate showed reduced pathogenicity compared to the parental virulent strain and induced protective immunity in vaccinated domestic pigs, although some mild clinical signs were observed [13]. 

In this manuscript, a detailed analysis of haematological parameters and inflammatory and immunological markers was conducted over time to deeply understand both the impact of Lv17/WB/Rie1 and its mutant Lv17/WB/Rie1/d110-11L on domestic pigs’ health status and their ability to protect against challenge with a virulent ASFV isolate. An improved understanding of virus–host interactions is required to generate safe and efficient ASFV vaccines, and to date, the mechanisms underlying the protection against virulent ASFV remain largely unknown [14,15]. 

Infection with Armenia/07 of the ASF virus resulted in inflammation, identified by high CRP concentration in sera, in agreement with a previous study with both genotypes I and II [20,24,25]. Our data revealed that the vaccination of pigs with both Lv17/WB/Rie1 and Lv17/WB/Rie1/d110-11L led to increased levels of this inflammation marker, although with lower intensity for the deleted mutant. After the challenge infection with Armenia/07 of the ASF virus, inflammation was of lower intensity in the vaccinated group, especially in the Lv17/WB/Rie1/d110-11L group. Platelet and RBC parameters observed for the virulent Armenia/07 of ASFV are similar to those described for other virulent strains belonging to both genotype I and genotype II isolates [20,26]. In particular, Walczak and colleagues reported that infection with the virulent Polish strain Pol18_28298_O111 (genotype II) resulted in decreased platelet number, RBC number, and HGB [20]. Our study described that the injection with non-HAD Lv17/WB/Rie1 led to a significant decrease in both platelet and RBC numbers and HGB. In contrast, the vaccination with the mutant Lv17/WB/Rie1/d110-11L did not alter domestic pigs’ haematological parameters (with the exception of a small reduction in platelet values at 7 dpc). It is very interesting to note that both the Lv17/WB/Rie1 strain and its derived mutant Lv17/WB/Rie1/d110-11L prevented the decline in platelet number, RBC number, HGB, and HCT post-challenge with Armenia/07 of the ASF virus. Overall, haematological data highlight the harmlessness of Lv17/WB/Rie1/d110-11L and its ability to mitigate the adverse reactions triggered by infection with the virulent isolate.

Challenge infection with Armenia/07 of the ASF virus also resulted in lymphopenia, in agreement to Hühr and colleagues. The researchers observed that infection with Armenia/08 of the ASF virus led to lymphopenia, whereas it did not alter levels of leukocytes, monocytes, and granulocytes [27]. In this study, we observed that the non-HAD Lv17/WB/Rie1 induced lymphopenia, and three out of five pigs presented lymphocyte levels below the reference range. Similar to what was observed for the virulent strain—Armenia/08—of ASF virus, IM administration of the non-HAD Lv17/WB/Rie1 resulted in decreased lymphocyte number and did not alter circulating levels of granulocytes or monocytes. A decrease in circulating lymphocyte levels might be due to apoptosis, which triggered the levels of pro-inflammatory cytokines observed in the early phase of post-ASFV infection, as previously speculated [26,28]. No decrease in monocyte number was observed, in agreement with two recent studies [20,27]. These findings might be explained by the tropism of ASFV to myeloid cells and its ability to prevent apoptosis in its target cells [29]. The three survivors in the Lv17/WB/Rie1 group presented normal lymphocyte levels at the time of the challenge infection, and the vaccination inhibited the appearance of lymphopenia induced by Armenia/07 of the ASF virus. It is very interesting to note that the attenuated deleted mutant did not induce lymphopenia and prevented its appearance post-challenge with virulent ASFV. 

Overall, these data support the harmlessness of Lv17/WB/Rie1/d110-11L and its ability to mitigate the adverse reaction of Armenia/07 of the ASF virus but revealed that its parental Lv17/WB/Rie1 presents significant clinical symptoms in some of the tested vaccinated pigs. Recent studies in wild boar instead showed the relative safety of this non-HAD strain, with orally vaccinated wild boar presenting only transient fever after immunization with a high dose (10^4^ TCID_50_) [30] and with an 83.33% survival rate after repeated vaccination with high doses of Lv17/WB/Rie1 (two doses of 10^4^ TCID_50,_ 18 days apart) [12]. Differences between domestic pigs and wild boars may be linked to differences in their immune system responses. Huhr and colleagues showed that the virulent genotype II strain induced lymphopenia in domestic pigs but not in wild boars. We might speculate that these differences also exist for Lv17/WB/Rie1, and conducting further studies would be interesting, to understand the impact of this non-HAD strain on the immune system of domestic pigs and wild boar [27].

Then, circulating levels of several pro-inflammatory cytokines were investigated. Infection with virulent ASFV strains belonging to either genotype I or II often results in increased levels of serum pro-inflammatory cytokines [28,31]. Our data revealed that vaccinating pigs with Lv17/WB/Rie1, but not Lv17/WB/Rie1/d110-11L, led to increased serum levels of four pro-inflammatory cytokines, suggesting that inflammation was lower in pigs vaccinated with the deleted mutant compared to the parent wild-type strain. After the challenge infection with Armenia/07 of the ASF virus, levels of pro-inflammatory cytokines were lower in pigs immunized with Lv17/WB/Rie1/d110-11L. Overall, these data support the harmlessness of this candidate vaccine and its ability to mitigate the inflammatory response triggered by Armenia/07 of the ASF virus.

Levels of two anti-inflammatory cytokines were monitored next. Vaccination with Lv17/WB/Rie1 and its deleted mutant led to increased levels of IL-1Ra, although with lower intensity for the latter. IL-1Ra levels were recently monitored in two other studies, where scientists observed its increase also in pigs infected with virulent SY18 of the ASF virus [32] or Armenia/08 the ASF strain [33], reaching the highest level on the day before death. Radulovic and colleagues also reported that serum levels of this receptor antagonist increased in pigs infected with moderately virulent isolate Estonia/2014, but with differences among animals with different immunological and hygienic status. Specific Pathogen-Free (SPF) pigs released higher levels of IL-1Ra at earlier stages post-infection than farm pigs, which was concomitant with lower circulating levels of pro-inflammatory cytokines and reduced mortality [33]. In our study, we observed that serum levels of IL-1Ra mirror those of CRP levels and also displayed a trend similar to IL-1α and IL-1β of the Lv17/WB/Rie1 immunization. Instead, Lv17/WB/Rie1/d110-11L-vaccinated pigs presented higher circulating levels of IL-1Ra but no elevated levels of pro-inflammatory cytokines. In this contest, IL-1Ra release seems to contribute to tuning down inflammation in vaccinated pigs, preventing the development of exacerbated immune responses. This cytokine might play an important role in ASFV immune-pathology [34]; thus, future studies should include IL-1Ra in the panel of circulating cytokines that should be monitored during ASFV experimental studies. 

Our data showed only a modest IL-10 increase in pigs vaccinated with Lv17/WB/Rie1 compared to controls at both 7 and 14 dpc, although without statistical significance. Nevertheless, two of the three tested subject presenting higher levels of IL-10 at 14 dpv died at 19 dpv. No increase in IL-10 sera levels was observed for its deleted mutant at any tested time point. In the past, two studies reported no significant changes in IL-10 levels after immunization or challenge infections with different ASFV strains [35,36]. Nevertheless, more recently, other studies described a negative correlation between the secretion of IL-10 and survival: increased IL-10 levels were observed in wild boar or pigs that later died of ASFV infection [31,37,38]. These studies suggest that the occurrence of IL-10 is not part of a physiologically orchestrated immune response but rather a sign of a fatally derailed system that will not recover, as recently reviewed [14,34]. 

There are currently conflicting results regarding the circulating levels of IFN-β in pigs infected with ASFV [34]. Some researchers observed increased levels of this cytokine after infection with high doses of virulent ASFV isolates [39,40]. On the contrary, it was recently described that infection with attenuated ASFV-Δ7R, but not its virulent parental strain, increased serum levels of IFN-β (at 6 dpc) [41]. Circulating levels of this cytokine were also investigated in this study and our data are in agreement with Li and colleagues because increased levels of this cytokine were observed in pigs vaccinated with Lv17/WB/Rie1/d110-11L but not its parental strain. Challenge with Armenia/07 of ASFV did not increase IFN-β levels, which was observed after the challenge infection with other virulent genotypes of ASFV [39,40]. This might be due to the different infection doses injected in pigs in the studies (10^2^ in our study and 10^4^ in the other two mentioned investigations) or because at a later time post-challenge, few control pigs survived and thus were impossible to test. Vaccinations of Lv17/WB/Rie1 or Lv17/WB/Rie1/d110-11L did not result in increased serum levels of IFN-γ. Although serum levels of IFN type II were not elevated in vaccinated pigs, three out of five pigs vaccinated with Lv17/WB/Rie1 virus and all pigs vaccinated with the deleted mutant survived infection with the ASFV Armenia/07. These data are consistent with what has recently been reviewed by Schäfer and colleagues: IFN-γ secretion is not a useful correlate of protection, as animals with high IFN-γ levels often still succumb to disease, and increased levels of this cytokine could also be the result of a derailed immune response in moribund animals [14].

Finally, no modulation of IL-4, CXCL8, or GM-CSF was observed either post-vaccination or post-challenge, in agreement with previous studies on virulent or attenuated ASFV strains [32,35,42].

## 5. Conclusions

In this study, the effects of Lv17/WB/Rie1 and the derived Lv17/WB/Rie1/d110-11L mutant on hematological and immunological parameters of domestic pigs were characterized in detail. We observed that vaccination with Lv17/WB/Rie1 presented some side effects, such as inflammatory response (characterized by increased serum levels of pro-inflammatory cytokines and CRP), and although transitory, it negatively affected some hematological parameters (decline in platelet levels, RBC parameters, lymphocytes numbers). Contrastingly, Lv17/WB/Rie1/d110-11L triggered a milder inflammatory reaction, with only a transitory increase in CRP levels, and induced a transitory drop of platelet levels. Interestingly, Lv17/WB/Rie1/d110-11L-vaccinated pigs presented high values of IL-1Ra without the increase in circulating levels of pro-inflammatory cytokines.

Both strains were able to counteract several adverse reactions elicited by challenge with Armenia/07, such as lymphopenia, decline in platelet number, RBC number, HGB, HCT, and inflammation (identified by high CRP concentration in sera). 

We previously described that both strains are unsuitable to be used as vaccines in their current forms [13], although Lv17/WB/Rie1/d110-11L triggered milder side effects compared to its parental strain Lv17/WB/Rie1. In this study, we conducted further in-depth investigations to better understand the differences between these strains and their impact on animals’ health and immune system. We believe that our work will help to better understand the immune–pathological mechanisms of ASF and can contribute to designing an efficient vaccine against this disease.

## Figures and Tables

**Figure 1 vaccines-11-01277-f001:**
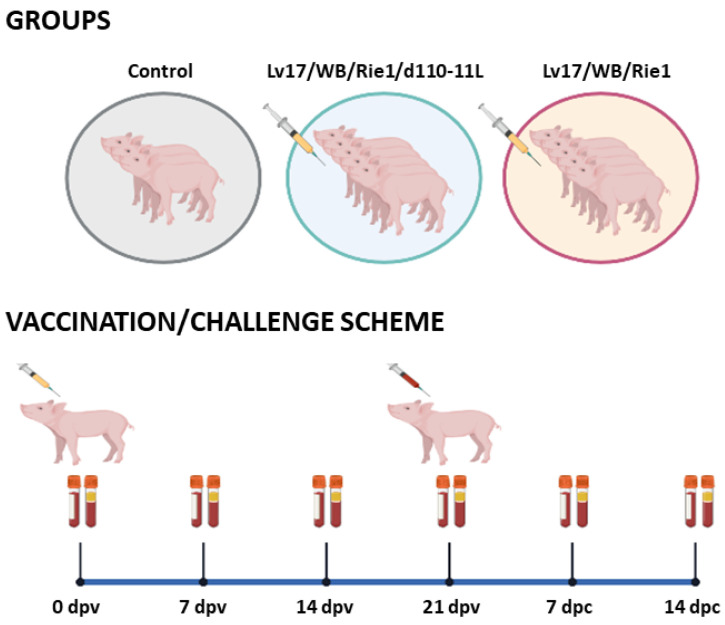
Vaccination and subsequent ASFV challenge infection. Schematic outline of the vaccine/challenge study highlighting the key time points during the entire experimental period. Animals were immunized with Lv17/WB/Rie1 (*n* = 5) or its derived mutant Lv17/WB/Rie1/d110-11L (*n* = 5), alongside a control group (*n* = 3). At 21 days post-vaccination (dpv), all animals were challenged with the virulent Armenia/07 of the ASF virus [13]. Blood samples were collected at 0, 7, 14, 21 dpv, and 7, 14 days post-challenge (dpc). Created with Biorender.com (accessed on 29 June 2023).

**Figure 2 vaccines-11-01277-f002:**
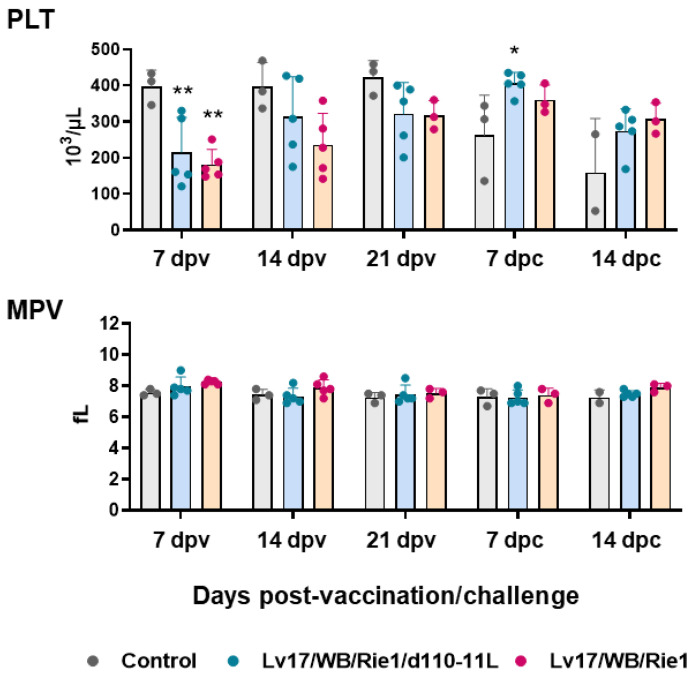
Platelets parameters in vaccinated and control pigs. Changes in the levels of platelet (PLT) and mean platelet volume (MPV) were monitored throughout the study. Data from three (controls, grey) or five (vaccinated with Lv17/WB/Rie1/d110-11L, quite blue) or five (vaccinated with Lv17/WB/Rie1 WT, orange) different pigs are shown. At each time point, values of vaccinated pigs (Lv17/WB/Rie1/d110-11L or Lv17/WB/Rie1 WT) were compared to controls; * *p* < 0.05, ** *p* < 0.01.

**Figure 3 vaccines-11-01277-f003:**
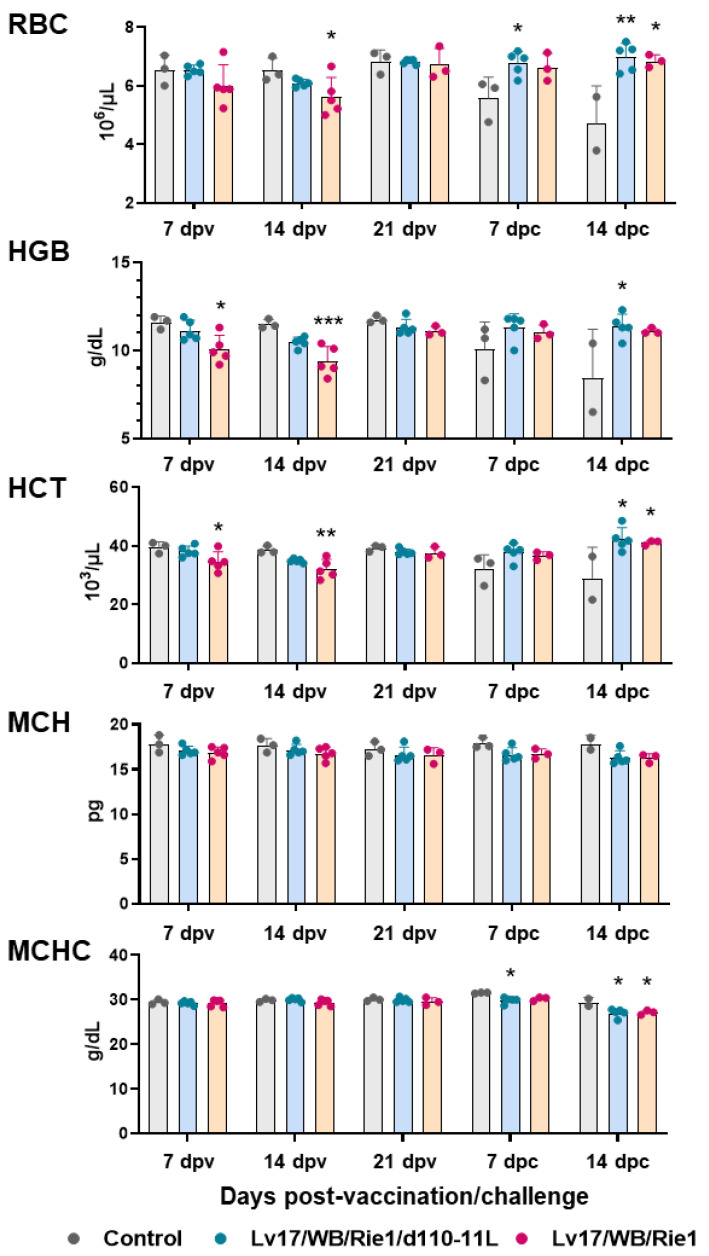
Red blood cells parameters in vaccinated and control pigs. Changes in the levels of red blood cells (RBC), hemoglobin (HGB), hematocrit (HCT), mean corpuscular haemoglobin (MCH) and its concentration (MCHC) were monitored throughout the experiment. Data from three (controls, grey) or five (vaccinated with Lv17/WB/Rie1d110-11L, quite blue) or five (vaccinated with Lv17/WB/Rie1, orange) different pigs are shown. At each time point, values of vaccinated pigs (Lv17/WB/Rie1/d110-11L or Lv17/WB/Rie1) were compared to controls; * *p* < 0.05, ** *p* < 0.01, *** *p* < 0.001.

**Figure 4 vaccines-11-01277-f004:**
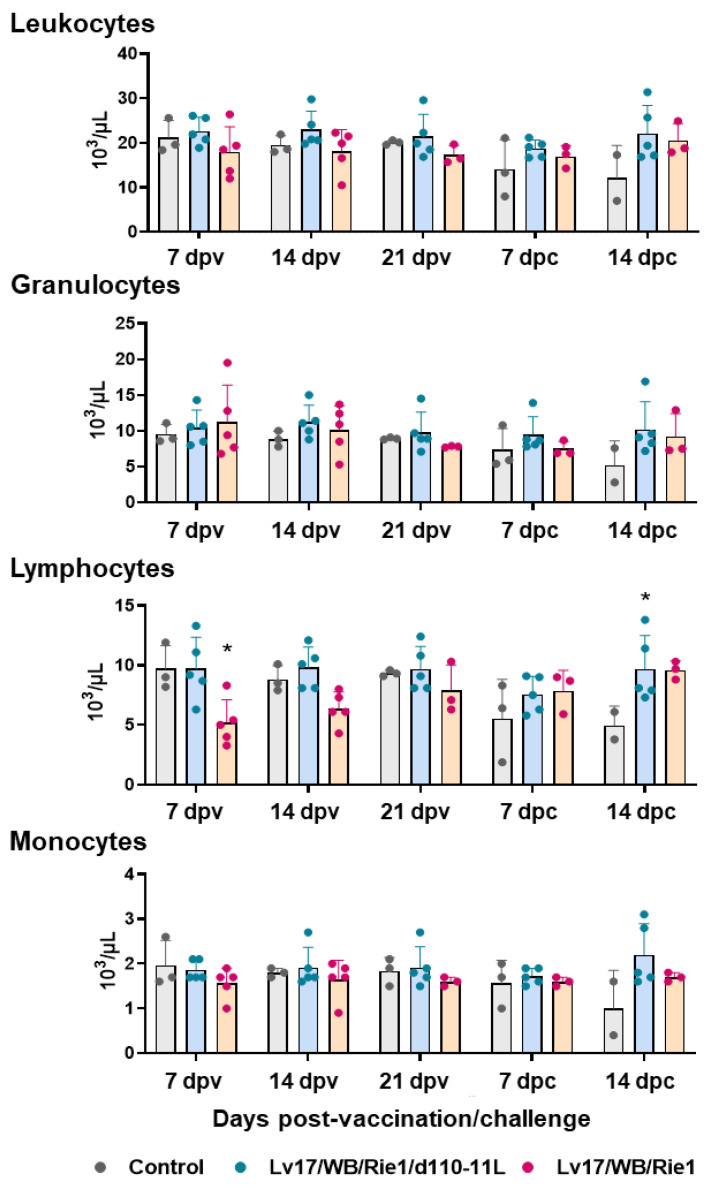
Circulating leukocytes levels in vaccinated and control pigs. Changes in the levels of circulating leukocytes, then divided into granulocytes, lymphocytes, and monocytes were monitored throughout the experiment. Data from three (controls, grey) or five (vaccinated with Lv17/WB/Rie1d110-11L, quite blue) or five (vaccinated with Lv17/WB/Rie1, orange) different pigs are shown. At each time point, values of vaccinated pigs (Lv17/WB/Rie1/d110-11L or Lv17/WB/Rie1) were compared to controls; * *p* < 0.05.

**Figure 5 vaccines-11-01277-f005:**
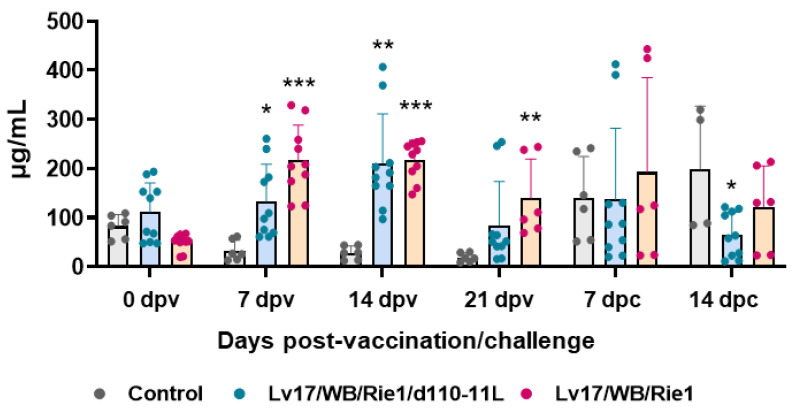
Kinetics of C-reactive protein levels in sera samples taken throughout the experiment. Sera samples were taken through the experiment and C-reactive protein levels were monitored though ELISA. Data from three (controls, grey) or five (vaccinated with Lv17/WB/Rie1/d110-11L, quite blue) or five (vaccinated with Lv17/WB/Rie1, orange) different pigs are shown. At each time point, values of vaccinated pigs (Lv17/WB/Rie1/d110-11L or Lv17/WB/Rie1) were compared to controls; * *p* < 0.05, ** *p* < 0.01, *** *p* < 0.001.

**Figure 6 vaccines-11-01277-f006:**
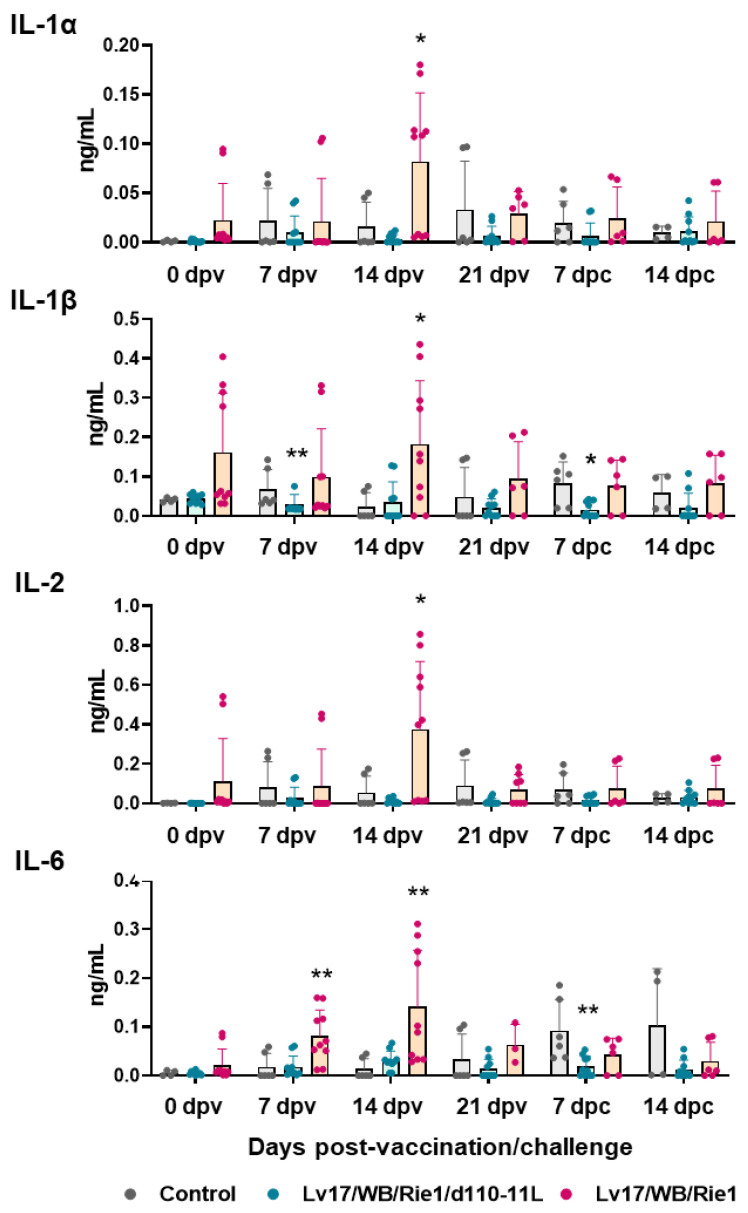
Serum pro-inflammatory cytokines levels in vaccinated and control pigs. Sera samples were taken throughout the experiment and their levels of IL-1α, IL-1β, IL-2, IL-6 were monitored though ELISA. Data from three (controls, grey) or five (vaccinated with Lv17/WB/Rie1/d110-11L, quite blue) or five (vaccinated with Lv17/WB/Rie1, orange) different pigs are shown. At each time point, values of immunized pigs (Lv17/WB/Rie1d110-11L or Lv17/WB/Rie1) were compared to controls; * *p* < 0.05, ** *p* < 0.01.

**Figure 7 vaccines-11-01277-f007:**
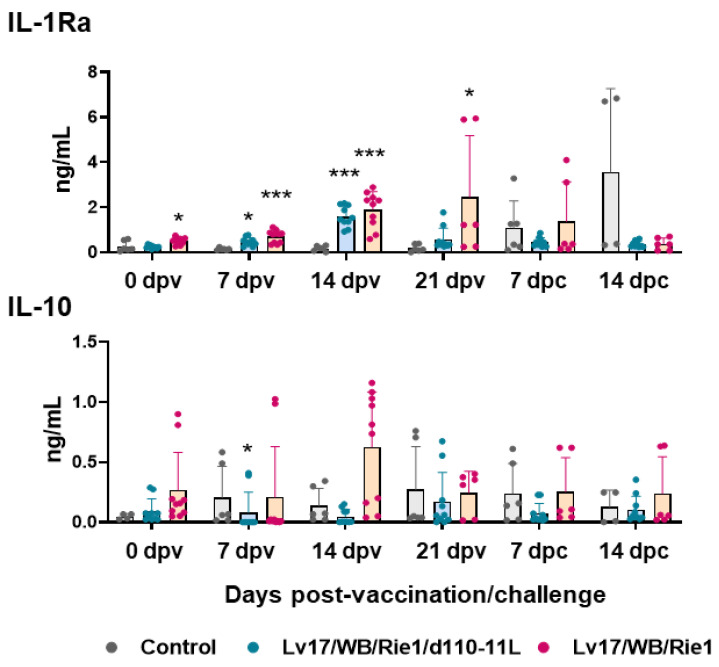
Kinetics of IL-1Ra and IL-10 levels in sera samples of vaccinated and control pigs. Sera samples were taken throughout the experiment and changes in the circulating levels of IL-1Ra and IL-10 were monitored though ELISA. Data from three (controls, grey) or five (vaccinated with Lv17/WB/Rie1/d110-11L, quite blue) or five (vaccinated with Lv17/WB/Rie1, orange) different pigs are shown. At each time point, values of immunized pigs (Lv17/WB/Rie1/d110-11L or Lv17/WB/Rie1) were compared to controls; * *p* < 0.05, *** *p* < 0.001.

**Figure 8 vaccines-11-01277-f008:**
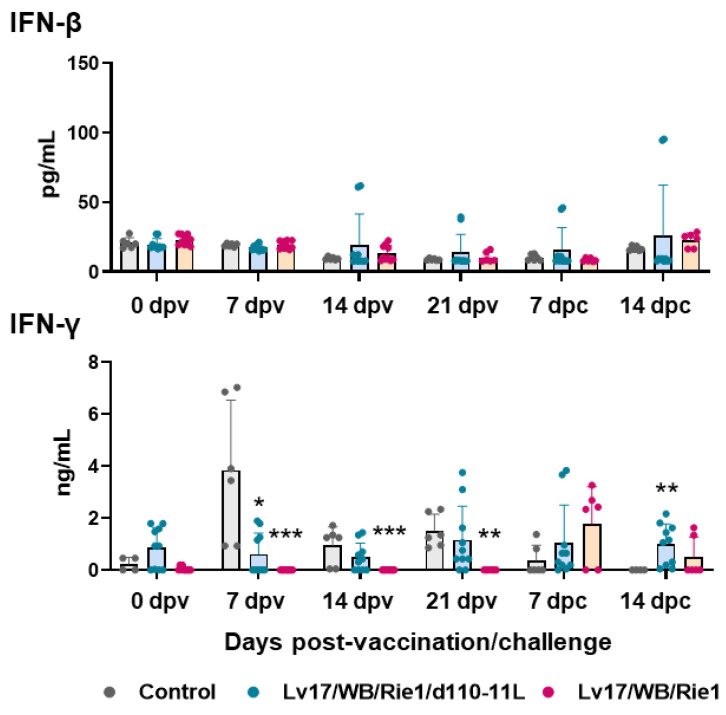
Serum levels of type I and II IFNs in vaccinated and control pigs. Sera samples were taken throughout the experiment and changes in the circulating levels of IFN-β and IFN-γ were monitored though ELISA. Data from three (controls, grey) or five (vaccinated with Lv17/WB/Rie1/d110-11L, quite blue) or five (vaccinated with Lv17/WB/Rie1, orange) different pigs are shown. At each time point, values of immunized pigs (Lv17/WB/Rie1/d110-11L or Lv17/WB/Rie1) were compared to controls; * *p* < 0.05, ** *p* < 0.01, *** *p* < 0.001.

**Table 1 vaccines-11-01277-t001:** List of haematological parameters evaluated during the study, indicated as abbreviation (unit of measure) and laboratory reference ranges in healthy pigs.

Parameters (Abbreviation)	Unit of Measure	Reference Range	Reference
White blood cells (WBC)	[10^3^/μL]	11–22	[17,18]
Total number of granulocytes	[10^3^/μL]	3.2–13.2	[17]
Total number of lymphocytes	[10^3^/μL]	4.5–13	[17]
Total number of monocytes	[10^3^/μL]	0.2–2	[17]
Platelets	[10^3^/μL]	100–900	[17]
Mean platelet volume (MPV)	fL	6.71–9.91	[19]
Red blood cells (RBC)	[10^6^/μL]	5–8	[17,18]
Hematocrit (HCT)	%	32–50	[17,18]
Hemoglobin (HGB)	[g/dL]	10–16	[17,18]
Mean corpuscular hemoglobin (MCH)	[pg]	17–21	[17]
Mean corpuscular hemoglobin concentration (MCHC)	[g/dL]	30–34	[17,18]

Unit of measures: 10^3^/µL, 10^3^ per microliter; fL, femtoliter; 10^6^/µL, 10^6^ per microliter; %, percentage; g/dL, grams per deciliter; pg, picogram.

## Data Availability

Not applicable.

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
