# Peer review of "Evaluation of Haematological and Immunological Parameters of the ASFV Lv17/WB/Rie1 Strain and Its Derived Mutant Lv17/WB/Rie1/d110-11L against ASFV Challenge Infection in Domestic Pigs"

_vaccines, 2023, doi:10.3390/vaccines11071277_

Round 1
Reviewer 1 Report
The manuscript submitted by Franzoni et al. entitled "Evaluation of haematological and immunological parameters of the ASFV Lv17/WB/Rie1 strain and its derived mutant Lv17/WB/Rie1/d110-11L against ASFV challenge infection in domestic pigs" aims to compare in deep several biological and biochemical blood parameters between swines infected with SFV Lv17/WB/Rie1 strain and the previously reported mutant Lv17/WB/Rie1/d110-11L. The manuscript is very well-written and brings a very good collection of new results that helps to characterize the vaccine properties of a new ASFV vaccine candidate.
In the opinion of this reviewer few points need to be improved before the acceptance of this manuscrit. Thus, only the following points need to be improved:
1) Line 47, a recent review about modern ASFV vaccines should be added (e.g. 10.1080/22221751.2022.2108342)
2) Along the manuscript the authors state that their results "highlight the safety of Lv17/WB/Rie1/d110-11L". Considering that long-time experiments are missing (more than 30 days after challenge) as well as viral shedding experiments, authors should replace the word "safety" by another one like "harmlessness"
3) Finally, the discussion section can be easly shorten since many info is repeat in the Introdution section
Reviewer 2 Report
This study investigated the haematological and immunological parameters of the ASFVLv17/WB/Rie1 strain and its derived mutant Lv17/WB/Rie1/d110-11L against ASFV challenge infection in pigs. The results showed that Lv17/WB/Rie1 triggered an inflammatory response, with increased levels of CRP and pro-inflammatory cytokines, and induced lymphopenia, thrombocytopenia and decline of red blood cells (RBC) parameters, although transitory. Lv17/WB/Rie1/d110-11L triggered only transitory thrombocytopenia and a mild inflammatory reaction, with no increase in serum levels of pro-inflammatory cytokines, but raise of IL-1Ra. Both strains, counteracted several adverse reactions elicited by virulent challenge, like thrombocytopenia, a decline in RBC parameters, and inflammation. The authors provided a deep portray of the impact of diverse ASFV strains on the domestic pig’s immune system. But the sample size is small. My comments are as follows:
Table 1: Please list the references from which the reference range is derived.
Figure 2, 3, and 4: Days post-vaccination/challenge in the X axis did not clearly marked.
Figure 6 and 7: What does it mean that some of the significantly different indicators also vary widely within the groups? Discussion
Line 103-106: Please re-introduce the sex, age, weight and genetic background of these experimental pigs. Because the number of pigs in this study was so small, individual information about the pigs was important to evaluate the trial results.
Line 159-164: an ANOVA was used in this study. The authors did not consider the effects of body weight, age, sex, genetic background of pigs. With a sample size of just a few pigs in each group, it's worth wondering how much confidence these results have.
Line 42: change "and recent outbreaks also been reported also" to “and recent outbreaks have also been reported”
Line 52: change "Nowadays, several groups tried" to “Nowadays, several groups have tried”
Line 59: Change "A non-haemadsorbing (non-HAD) ASFV strains" to “A non-haemadsorbing (non-HAD) ASFV strain”
Line 68: "this non-HAD strains was" to “this non-HAD strain was”
Line 190: “IM administration of Lv17/WB/Rie1 lead to a decrease” to “IM administration of Lv17/WB/Rie1 led to a decrease”
Line 210: “A total number of leukocytes” to “The total number of leukocytes”
Line 215: “the control group were tested 14 dpc” to “the control group were tested at 14 dpc”
Round 2
Reviewer 2 Report
Table 1: Please list the references from which the reference range is derived. Please mark the relevant references for these values in Table 1, not just in the text. This makes it easier for the readers to read without having to look it up in the text.
Please unify the label of key time points on the X-axis in Figure 2 to Figure 8. Also, I recommend keeping the font size consistent across the Figures.
Line 66 and 68: change "[10-]" to "[10]"
Line 123: delete a period. change to "the experiments. All animals"
